# Unravelling the 2e^−^ ORR Activity Induced by Distance Effect on Main-Group Metal InN_4_ Surface Based on First Principles

**DOI:** 10.3390/molecules27227720

**Published:** 2022-11-09

**Authors:** Peng Li, Jiawen Xu, Yaqiong Su

**Affiliations:** 1School of Chemistry, Xi’an Key Laboratory of Sustainable Energy Materials Chemistry, State Key Laboratory of Electrical Insulation and Power Equipment, Engineering Research Center of Energy Storage Materials and Devices of Ministry of Education, Xi’an Jiaotong University, Xi’an 710049, China; 2General Research Institute of Engineering of Gotion High-Tech, Baohe District, Hefei 230041, China

**Keywords:** distance effect, 2e^−^ ORR, main group metal, metal-N doped graphene

## Abstract

The p-electron-dominated main-group metals (Sb, Se, In, etc.) have recently been reported to possess excellent oxygen reduction reaction (ORR) activity by means of heteroatom doping into graphene. However, on these main group metal surfaces, other approaches especially the distance effect to modulate catalytic activity are rarely involved. In this work, the origin of excellent 2e^−^ ORR catalytic activity of graphene-supported InN_4_ moiety by tuning the distance between metallic In atoms is thoroughly investigated by employing the first-principles calculations. Our DFT calculations show that the 2e^−^ ORR catalytic activity strongly depends on the crystal orbital Hamilton population (COHP) between In and O atoms. This work is useful for the rational design of main group metal single atom electrocatalysts.

## 1. Introduction

The oxygen reduction reaction (ORR) is an extremely important electrocatalytic process for energy storage and energy conversion [1]. So the ORR activities of many materials have been reported in the literature [2,3]. Among them, single atom catalysts (SACs) have been reported frequently in recent years, especially metal-N doped carbon nanomaterials [4,5,6]. Transition metal catalysts have an irreplaceable position in catalyst design due to their unique d orbital, and the d-band center model is widely used to understand and describe their ORR activity. Unlike transition metals, main-group metals have very wide sp-band and inactive d electrons that are considered to be of low activity in catalyst design [7]. However, main-group metal atoms anchored on nitrogen-modified graphene with non-delocalized sp-bands (such as Sb, Se, In) have been recently reported to exhibit promising 2e^−^ ORR catalytic activity [8,9,10,11]. In previous studies, the modified graphene by transition metal atoms (Fe, Co, Ni, Cu, Mn, etc.) and adjacent non-metallic atoms (N, B, P, S, etc.) and the morphology of carbon materials had been reported to effectively modulate the ORR catalytic activity [12,13,14,15,16,17,18]. And the metal loading (distance between adjoining metal atoms) may play an important role in adjusting this [19,20,21,22]. However, the distance effect of main-group metal atoms supported on nitrogen-modified graphene has not been reported.

In this work, the p-electron-dominated 2e^−^ ORR on the In and N co-doped graphene is systematically investigated by means of density functional theory (DFT) calculations. The distance between In atoms is controlled by adjusting the supercell size, which is a commonly used method for adjusting the distance (atomic ratio) in theoretical studies. We found that the InN4 site of (4 × 4) graphene supercell has the optimal catalytic activity of 2e^−^ ORR. The density of states (DOS) and crystal orbital Hamilton population (COHP) results well explain the correlation of the 2e^−^ ORR catalytic activity with the distance between In atoms.

## 2. Results

### 2.1. Calculation Process

The distance of In atoms is controlled by the supercell size as shown in Figure 1a–c and Appendix A [22,23]. As is shown in Appendix A, the doping energy of In atom (*E*_doping_) are all smaller than −3.40 eV, implying the stable formation of In atom on graphene. *E*_doping_ is defined as *E*_doping_ = *E*_InN4_ − *E*_NG_ − *E*_In_, where *E*_InN4_, *E*_NG,_ and *E*_In_ are the energy of the InN_4_, N-doped graphene (NG), and In atom [23]. In addition, we also investigated the In adsorption on graphene as shown in Appendix A, which shows the unstable In adsorption compared to the formation of InN_4_, indicating unfavorable diffusion of the In of InN_4_ to the graphene surface. The In atom is suspended on the surface of the InN_4_-doped graphene at a height of about 1.403, 1.410, and 1.404 Å in (3 × 3), (4 × 4), and (5 × 5) supercells, respectively, as shown in Figure 1d. Here we calculate the ORR process of 2e^−^ and 4e^−^ under acidic conditions (pH = 1). The specific reaction process can be represented by the following six step-by-step reactions: O_2_ + * → *O_2_(1)
*O_2_ + H^+^ + e^−^ → HOO*(2)
HOO* + H^+^ + e^−^ → O* + H_2_O(3)
O* + H^+^ + e^−^ → HO*(4)
HO* + H^+^ + e^−^ → H_2_O(5)
HOO* + H^+^ + e^−^ → H_2_O_2_(6)
in the above reaction formula, * is the surface of the catalyst material, and HOO*, O*, HO* are the three intermediates involved in catalysis. Reactions 1 to 5 are 4e^−^ processes, and reactions 1, 2, and 6 are 2e^−^ processes. Their reaction Gibbs free energies can be approximately calculated according to the following formula:ΔG = ΔE + ΔZPE − TΔS + ΔG_U_ + ΔG_pH_(7)
U_over_ = |U_0_ − U_L_|(8)
where E is the theoretical internal energy of the DFT calculation, ZPE is the zero-point energy, and ΔS is the entropy change. G_U_ = eU, where U is the electrode potential, G_pH_ = k_B_Tln × pH, and in this work, pH = 1 is taken into account. The overpotential is calculated by Equation (8). U_over_, U_0,_ and U_L_ are the overpotential, standard electrode potential, and limit the potential of a catalytic reaction, respectively.

### 2.2. ORR Catalytic Activity

The potential energy profiles of 2e^−^ and 4e^−^ ORR by InN_4_ sites are depicted in Figure 2c, and the results demonstrates that the 4e^−^ ORR process takes place much more difficult than 2e^−^ ORR. We also examined the various distances between In atoms for 2e^−^ and 4e^−^ ORR, and as shown in Figure 2a,b, and the 4e^−^ ORR is always suppressed by 2e^−^ ORR. There is a linear relationship between the OOH adsorption energy (ΔG_OOH_) and U_over_ (U_over_ = |ΔG_OOH_ − 4.24|) as shown in Figure 2a. Therefore, in this work, we mainly analyze the regulation of the 2e^−^ ORR activity on InN_4_ sites by changing the distance between In atoms. Li’s group recently reported that InN_4_ sites exhibit the excellent 2e^−^ ORR activity, in line with our findings [11]. While we also found that the InN_4_ sites on the (4 × 4) graphene display the highest 2e^−^ ORR catalytic activity than the other candidates as shown in Figure 2a. On the InN_4_ sites of (3 × 6) graphene, the overpotential of 2e^−^ ORR is 0.74 V, while drastically lowered to 0.15 V on the InN_4_ sites of (4 × 4) graphene. Obviously, the distance between In atoms strongly influences the 2e^−^ ORR activity.

To explore the distance effect of In single atoms during 2e^−^ ORR, we conducted a thorough electronic structure analysis, including Bader charge [24], the density of states, and COHP. Unexpectedly, the Bader charge does not serve as a viable theoretical activity descriptor for distance effects on InN_4_ surfaces as shown in Appendix A. Previously, the partial density of states (PDOS) of d_z2_ and d_yz_ of supported transition metal atoms is generally used to correlate their ORR or carbon dioxide reduction reaction (CO_2_RR) activity [25,26]. In this work, the p orbitals of In atoms are examined. As shown in Appendix A, it’s clear from either the DOS or the band structure that the weight of p_z_ is dominant in the p orbitals. While the catalytic activity (overpotential) of 2e^−^ ORR does not have a positive correlation with the center of their p_z_ or p_y_ orbital energy levels as shown in Appendix A. We found that the overall p_z_ level distribution is somehow related to the 2e^−^ ORR catalytic activity. Compared with InN_4_-4×, the p_z_ orbital energy levels of InN_4_-5 × 6 and InN_4_-6 × 6 have a shift to the right, and InN_4_-5 × 5 seems to shift a little to the right compared to InN_4_-6 × 6. The reason is that the p_z_ orbital is the host-orbital in the InN_4_ system, and the higher the energy level of the p_z_ orbital, the higher the energy level of the antibonding orbital when the adsorbing small molecules such as OOH, which enhances the adsorption energy. As shown in Figure 3b, the p_z_ orbital energy levels of InN_4_-3 × 5 and InN_4_-3 × 4 are shifted to the Fermi level compared to InN_4_-4 × 5, which reveals that InN_4_-3 × 6 has extremely poor 2e^−^ ORR activity. Moreover, when the OOH is adsorbed on the surface of the InN_4_ site, the electrons of In atoms are transferred to the surroundings of O, H, and N atoms, especially the z direction, which is shown in Figure 3c and Appendix A. Conclusively, the p_z_ orbital energy level of the In atom can roughly describe the 2e^−^ ORR catalytic activity trend of the InN_4_ site.

In addition, as noted from the energy band result diagram, the Conduction Band Minimum (CMB) and Valence Band Maximum (VBM) was intersecting in the 3 × 6 and 3 × 4 supercells, which facilitated the electron transport between adsorbed molecules and In atoms, making the OOH desorption more difficult and detrimental to the overall ORR process. In view of this, we analyze the bond strength between the In atom and the O atom of OOH. The COHP can describe the covalent bond strength between, and understand the distribution of bonding and antibonding orbital energy levels [27]. By calculating the COHP between the In single atom and O atom, we find the p_z_ orbital is the main contributor to the In-O bond as shown in Figure 3d, which is consistent with the PDOS results. We found that ORR activity is related to the anti-bonding and bonding energy levels between In atoms and O of the OOH intermediate. The lower the antibonding or bonding energy level, the stronger the ORR catalytic activity, which is shown in Figure 4a–d and Appendix A. It can be understood that InN_4_-4 × 4 and InN_4_-6 × 6 have more anti-bonding states around the −4.0 eV energy level, and the anti-bonding energy level of InN_4_-4 × 4 is lower. The bonding energy level of InN_4_-5 × 6 is the lowest (about −0.5 eV), while the others are basically around 3.0 eV in this work. Obviously, the COHP can well explain the ORR activity of InN_4_ with different distances between the neighboring In atoms.

### 2.3. Theoretical Descriptor

Although the results of PDOS and COHP can explain their ORR activity, do not predict the ORR activity of the InN_4_ site well. Metal atomic spacing or loading ratio has recently been reported as an activity descriptor [19]. As shown in Appendix A, the distance and doping ratios of In atom cannot serve as good descriptors for ORR catalysis. While we found that the integral of Crystal Orbital Hamilton Population (ICOHP) between the 5 p_z_ or 5p orbital of In single atom and the O has a volcanic relationship with the 2e^−^ ORR overpotential as shown in Figure 4e,f. ICOHP is the integral of COHP, which can quantitatively describe the strength of the chemical bond. The larger the ICOHP, the stronger the chemical bond [28]. InN_4_-3 × 6 is located at the apex of the volcano, while InN_4_-4 × 4 and InN_4_-5 × 6 are located on both sides of the volcano curve. The reason is that the bonding energy level between the In atom and O atom of adsorbed OOH intermediate is shifted to the Fermi level (about −2.0 eV) in InN_4_-5 × 6, which lowers the value of the ICOHP, while the anti-bonding energy level between the In atom and O atom in InN_4_-4 × 4 is shifted below the Fermi level, and the overall −ICOHP is bigger. All in all, the more the bonding orbital energy level near −2.0 eV is shifted to the Fermi level, the smaller the −ICOHP between the 5 p_z_ or 5p orbital of the In atom and the O atom. If the antibonding orbital energy level is farther from the Fermi level, the bigger the −ICOHP. In this work, we propose a new descriptor for 2e^−^ ORR activity on the InN_4_ sites.

## 3. Discussion and Conclusions

In this work, based on first-principles calculations, we first calculated the ORR activity on the InN_4_ surface with the different In-In atom distances. We found that the 2e^−^ process is better than the 4e^−^ process, and the distance between In atoms can significantly influence the 2e^−^ ORR activity. The overpotential of the ORR on the surface of InN_4_ was 0.15–0.74 V by adjusting the distance of In-In in our DFT calculations. By calculating and analyzing the electronic structure, we found that the p electrons of In atom especially those in the p_z_ orbital play a very important role in the regulation of ORR catalytic activity. The p_z_ orbital energy level of the In atom from the DOS diagram could explain the order of ORR activity at InN_4_ sites. Moreover, we also found that the anti-bonding and the bonding energy level between In and the O atoms of OOH could explain their ORR activity by analyzing the covalent bond strength between the In and the O atom of OOH. However, the p-band energy levels and the COHP between the In and the O atom cannot be used as theoretical activity descriptors of ORR at InN_4_. After a series of attempts at the theoretical activity descriptors such as Bader charge, the metal atom load ratio of the In atom as well as the distance of In-In, we propose that the ICOHP between the p electrons of the In atoms and the surface adsorbed OOH intermediate can be used as a theoretical activity descriptor for ORR activity at InN_4_ sites. This work provides a theoretical understanding of the screening and designing of the main group metal catalysts in heterogeneous electro-catalytic reactions.

## 4. Materials and Methods

Calculations were performed under periodic boundary conditions employing Vienna Ab-initio Simulation Package (VASP). Projector-augmented wave (PAW) pseudopotentials and Perdew–Burke–Ernzerhof (PBE) exchange-correlation functional of generalized gradient approximation (GGA) were used to describe the interaction between core electrons [29,30]. C (2s, 2p), H (1s), O (2s, 2p), N (2s, 2p), and In (5s, 4p) electrons are investigated as valence electrons in our DFT calculations. The vaspsol implicit solvent model is introduced to describe solute-solvent interactions in density functional theory calculations [31,32,33]. And the DFT-D3 method with Becke-Johnson (BJ) damping is used to approximately describe the dispersion effect in the system [34]. In order to ignore the interlayer interactions, a vacuum layer of 15 Å was set up in the direction perpendicular to the 2D surface (z-directions). The cut-off energy was set to 400 eV and the convergence thresholds for the electronic structure and forces were set to 10^−4^ eV and 0.05 eV/Å, respectively. We have conducted convergence tests and found the energy difference of the 4 × 4 InN_4_ within 0.09 eV by using different convergence thresholds as shown in Appendix A. The geometric optimizations and DOS analysis were performed using a standard Monkhorst− Pack grid sampling at 3 × 3 × 1 and 11 × 11 × 1, respectively. After DFT calculations, the DOS and the free energy are calculated, based on the VASP toolkit (vaspkit) [35]. The COHP calculations and analysis are implemented in the Lobster program [36].

## Figures and Tables

**Figure 1 molecules-27-07720-f001:**
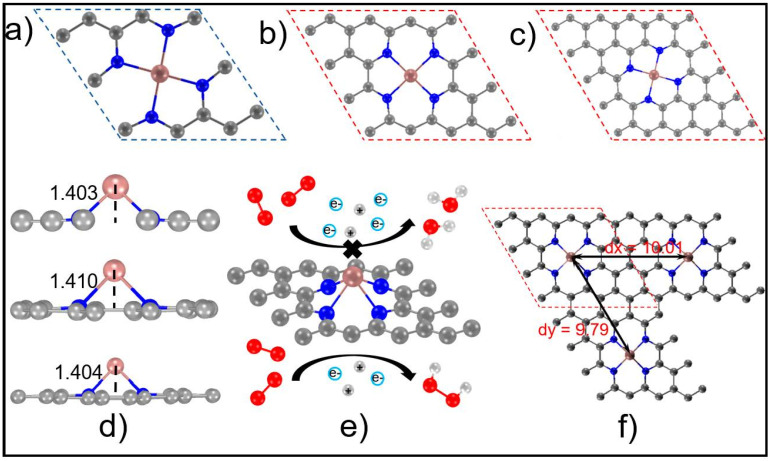
(**a**–**c**) The optimized structure of InN_4_ in different unit cells (from left to right are 3 × 3, 4 × 4, 5 × 5); (**d**) The distance that the In atoms on the surface of InN_4_ are offset from the two-dimensional plane (from top to bottom are 3 × 3, 4 × 4, 5 × 5); (**e**) the ORR process that occurs on the InN_4_ surface; (**f**) the distance between InN_4_ adjacent metal In atoms in 4 × 4 unit cell. The unit of distance in the figure is Angstrom (Å) and gray-black, off-white, red, blue, and maroon are C atoms, H atoms, O atoms, N atoms, In atoms, respectively.

**Figure 2 molecules-27-07720-f002:**
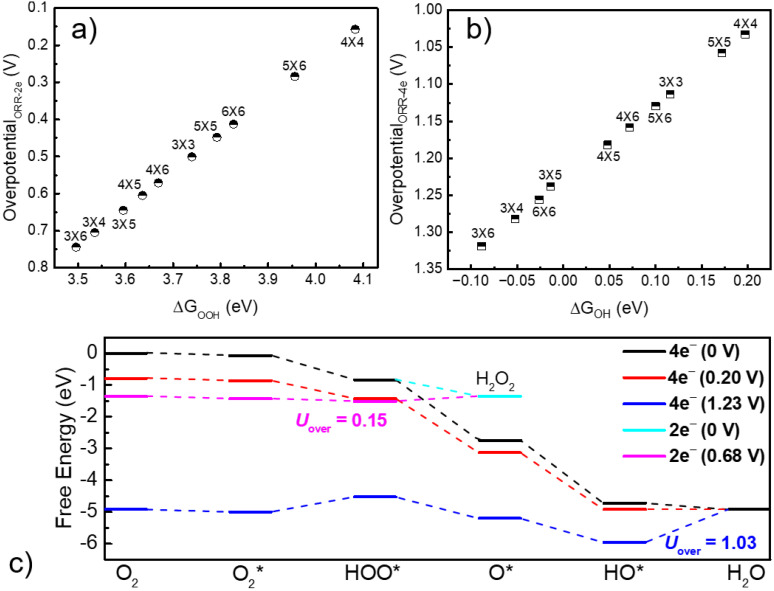
The catalytic activity of ORR on InN_4_ surfaces in different unit cells. (**a**) The linear relationship between 2e^−^ ORR catalytic activity and OOH adsorption free energy; (**b**) The linear relationship between 4e^−^ ORR catalytic activity and OH adsorption free energy; (**c**) The potential energy curve of ORR process on InN_4_ (4 × 4) surface.

**Figure 3 molecules-27-07720-f003:**
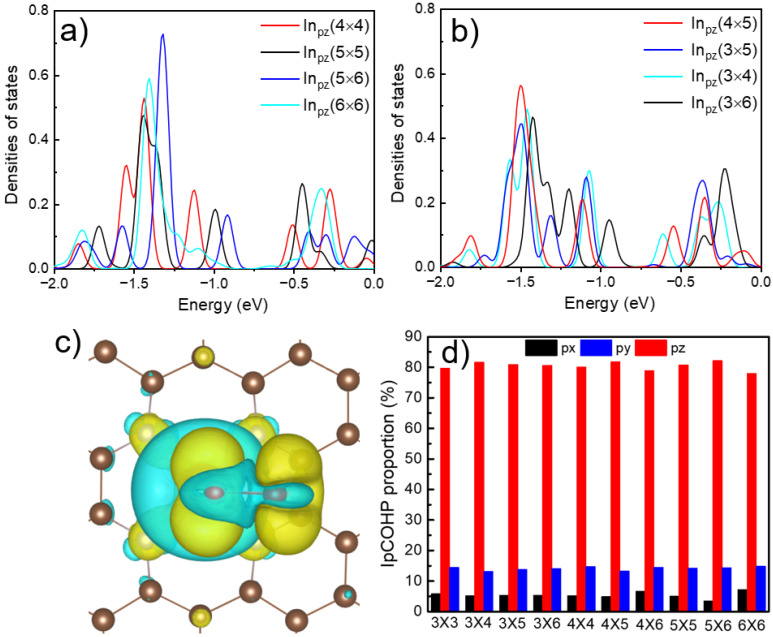
(**a**,**b**) The p_z_ orbital PDOS of InN_4_ is distributed in different unit cells. (**a**) several InN_4_ material with decent 2e^−^ ORR activity; (**b**) The PDOS distribution of the p_z_ orbital of less active InN_4_; (**c**) The difference charge distribution of the surface of InN_4_ (4 × 4) with adsorbing OOH. Yellow is the electron accumulation area and cyan is the electron deficient area; (**d**) The integral of the partial Crystal Orbital Hamilton Population (IpCOHP) proportion distribution between In atoms and O (OOH) atoms on the surface of InN_4_.

**Figure 4 molecules-27-07720-f004:**
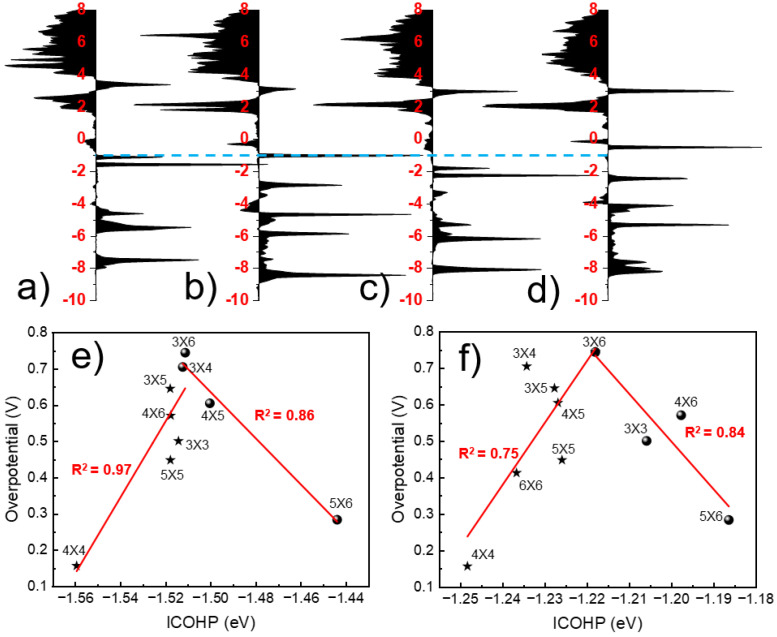
(**a**–**d**) The COHP distribution of In-O bonds in different supercells of InN_4_, and from left to right are InN_4_-3 × 3, InN_4_-4 × 4, InN_4_-5 × 6, InN_4_-6 × 6; (**e**) The volcano-like relationship between 2e^−^ ORR overpotential and the integral of the Crystal Orbital Hamilton Population (ICOHP) between 5p orbital of In atom and s and p orbitals of O (OOH); (**f**) The volcano curve between 2e^−^ ORR overpotential and the integral of the Crystal Orbital Hamilton Population (ICOHP) between the 5 p_z_ orbital of In atom and the s and p orbitals of O (OOH).

## Data Availability

The data presented in this study are available on request from the corresponding author.

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
