# Peer review of "Unravelling the 2e ORR Activity Induced by Distance Effect on Main-Group Metal InN4 Surface Based on First Principles"

_molecules, 2022, doi:10.3390/molecules27227720_

Round 1

Reviewer 1 Report

Comments to the Author:

The authors calculated the ORR activity of In single atoms of the graphene-supported InN4 fraction by adjusting the supercell of graphene, and their first-principles calculations revealed that pz orbital energy levels of In atoms can roughly describe the catalytic activity of 2e- ORR. By analyzing the strength of covalent bonds between In and O atoms, the authors found that the antibonding and bond energy levels between In and O atoms can also explain their ORR activity when OOH is adsorbed on the InN4 site. After a series of attempts on theoretical activity descriptors such as Bader charge, metal atom loading rate, and distance, the authors proposed that the Crystal Orbital Hamilton Population (ICOHP) between the p-electrons of In atoms and the adsorbed OOH intermediates on the surface can well describe the origin of 2e- ORR activity. Hopefully the authors could publish it after solving the following problems.

1. Usually, the size of supercell is the criterion used to judge whether it is reasonable to use it in calculations [J. Am. Chem. Soc. 2017, 139, 12480−12487]; if as authors stated “And the distance between In atoms is controlled by adjusting the supercell size, which is a commonly used method for adjusting the distance (atomic ratio) in theoretical studies”, related literature should be cited. 

2. “The cut- off energy was set to 400 eV and the convergence thresholds for the electronic structure and forces were set to 10−4 eV and 0.05 eV/Å, respectively.” This is a very rough accuracy, did the author do any testing calculations for reliability?

3. Is “px” in the Discussion and Conclusions “pz”? “px” is not mentioned in the main text. “By calculating and analyzing the electronic structure, we found that the p electrons play a very important role in ORR catalysis, especially those in the px orbital.”

4. Can the specific R^2 coefficient be given for the volcano diagram given in Figure 4.

5. The authors use the integral of Crystal Orbital Hamilton Population (ICOHP) as the activity descriptor of the overpotential. Please give the physical meaning of the integral of ICOHP specifically. 

6. The authors conclude that the distance between metals has an effect on ORR activity, does the magnetic variation between the two metals also have an effect on ORR activity? The authors are suggested to perform tests based on the recent article (Adv. Powder Mater., 2022, 1, 100031).

7. The authors might also calculate the kinetic energy barriers for different sizes of supercell.

8. The stability of the catalyst was not considered in the article, and the single-atom catalyst should consider whether the metal atoms diffuse on the substrate and calculate the corresponding transition states [J. Am. Chem. Soc. 2017, 139, 12480−12487].

9. From Fig. 2a, b it can be seen that the supercell symmetric structures 4×4, 5×5, 6×6, generally give better results than the asymmetric 3×6, 4×6, 5×6, etc. The authors should further analyze and discuss this phenomenon.

10. “The In atom is suspended on the surface of the InN4-doped graphene at a height of about 1.403, 1.410, and 1.404 Å in (3X3), (4X4) and (5X5) supercells, respectively as shown in Figure 1d. H”, the height of In suspended to the graphene plane is irregular regarding to the size of supercell?

11. Some format issues: “Crystal Orbital Hamilton Population (ICOHP)” is not necessary to appear for each time; “pz”, “dz2”, “dyz”, the component pf p or d orbits should be subscript.

Author Response

Please refer to the attached comments.

Reviewer 2 Report

This paper is on the computational work on the study of the oxygen reduction reaction for InN4-doped graphene configurations. The authors use density functional theory under the well-known VASP code for their calculation in addition to other software like vaspsol and Lobster. The paper seems scientifically sound. However, additional calculations and changes in the text are needed before it is accepted for publications (major revisions).

1)    The authors used VASP under PAW pseudopotentials. The authors need to include the valence electron configuration for each element that use these pseudopotentials.

2)    The authors need to include band structure calculation with projected orbital information in addition to their DOS.

3)    What is the reason of the term “excellent” in the title?

4)    What is the program used for the Bader calculated charges?

5)    The paper lacks strong conclusions and thus it is not clear to the reader what the authors achieved with this work. Thus, the section “Discussion and Conclusions” needs to be re-written.

6)    There are several grammatical errors throughout the text that need to be corrected. For example, in the abstract “p-electron dominated main-group metals” needs a “The” in front of this sentence. Also in lines 22-23 the sense should not start with “And.”

Author Response

Please refer to the attached comments.

Round 2

Reviewer 1 Report

The authors addressed all my concerns in the revised version, and I suggest it publication.

Reviewer 2 Report

The paper could now be accepted as is.